# OpenReview forum: "Debiasing Large Vision-Language Models by Ablating Protected Attribute Representations"
_NeurIPS.cc/2024/Workshop/SafeGenAi — SafeGenAi Poster_

### Official Review · Reviewer_U98Z · 2024-10-09

**Rating:** 6
**Confidence:** 3

**Review:**

## Summary of paper
The authors present a novel debiasing framework which does not require any extra training. From their experiments, they show that their method reduces the generation of protected attributes by over 50%.

## Strengths
- The proposed approach is effective at reducing the generation of the protected attributes.
- The proposed approach is training-freed, avoiding the high computational cost and data requirements typical of retraining approaches.
- The paper uses multiple evaluation methods, including bigram frequency matching, GPT-4o-based evaluations, and the DSL framework, to assess bias reduction.

## Weaknesses
- The method relies on defining biased attributes based on manual annotations and contrastive examples, which may introduce some subjectivity. What constitutes a biased attribute can vary across individuals or groups, and this subjectivity might affect the robustness of the method across different contexts.
- Because you are already using a list of words in order to reduce their generation, why not just use a block list at runtime instead? If one of the generated words is in the block list, the model has to regenerate or something along these lines.
- Why would race, age, etc. be protected in many of these instances? For example, in Figure 2, the man is Asian. Why is this protected? You are essentially just asking the model to include another fact about the image. Maybe a different example would make more sense here?
- The technique assumes that bias can be linearly represented and removed by ablating specific directions in the model’s residual stream. More complex forms of bias may not align with this assumption.

## Notes
- I am interested to read the longer version of this paper. This paper would benefit from more explanation, examples, etc.

---

### Official Review · Reviewer_Hm8c · 2024-10-09

**Rating:** 6
**Confidence:** 3

**Review:**

- Figure 2, right needs to be improved
- The evaluation relies on human-annotated methods like bigram frequency matching and GPT-4o, which are prone to subjectivity in determining what constitutes a bias-related reference. Because the method is currently optimized for race and appearance, this limits its applicability to other forms of bias like gender or age without further refinement.

---

### Official Review · Reviewer_DrEo · 2024-10-11

**Rating:** 6
**Confidence:** 4

**Review:**

This paper addresses an important and timely issue by proposing a novel debiasing framework for Large Vision Language Models (LVLMs). The authors tackle the challenge of societal biases embedded in training datasets and offer an efficient approach to ablate biased attributes during text generation without requiring retraining. The contribution is noteworthy for several reasons:

Innovative Approach: The proposed method of directly abating biased attributes in the text generation process is both creative and practical. By avoiding the need for extensive retraining and relying on a relatively small number of biased output samples (~1000), the approach demonstrates high efficiency and applicability.

Maintaining Model Performance: One of the key strengths of this work is that the debiasing is achieved without sacrificing performance. The authors demonstrate that the debiased model retains similar captioning performance on real-world datasets (e.g., COCO), which is a crucial outcome for ensuring practical usability.

Use of Synthetic Data: The successful incorporation of synthetic data to guide the debiasing process further highlights the flexibility of the approach. This could open up possibilities for applying the method across various datasets and tasks.

While the paper presents a strong contribution, there are a few areas where further improvements could be considered:

Scalability: While the method is effective with a small set of biased samples, it is unclear how well it scales with more complex or larger datasets that might exhibit more nuanced biases. It would be helpful to explore whether the technique can handle a broader range of biases beyond the ones addressed in the experiments.

Bias Definition and Generalization: The paper could benefit from a more detailed discussion on how "bias" is defined in different contexts. Biases can be context-dependent, and the current method might not generalize to all forms of demographic or societal biases. Expanding the scope to consider different types of biases and their interactions would strengthen the overall impact of the work.

Evaluation Metrics: While the authors show that accuracy is preserved post-debiasing, it would be valuable to include additional evaluation metrics, particularly those that measure fairness or bias reduction more explicitly. This would provide a more comprehensive picture of the method's effectiveness.

In summary, this paper offers a novel and efficient solution to a critical issue in LVLMs and presents a compelling set of experiments to validate its claims. With some additional attention to scalability, bias generalization, and evaluation metrics, it could have a significant impact on future work in this field.

---

### Official Review · Reviewer_Sr3N · 2024-10-12
**Effective Debiasing Method for LVLMs by removing protected attribute representations**

**Rating:** 7
**Confidence:** 4

**Review:**

**Summary**:
This paper presents a method for debiasing LVLMs by estimating and removing bias attribute directions from the models' internal representations. The experiments show that this approach effectively reduces the generation of biased or protected attributes.

**Strengths**:
1. Debiasing LVLMs is an important topic, and this work offers valuable insights with a simple and effective method.
2. The writing is clear, well-structured, and easy to follow.

**Weaknesses**:
1. More experiments are needed to confirm that the method does not negatively impact performance on general VQA tasks or degrade the quality of the generated text, even though it alters the internal representations of the models.
2. Comparisons with simpler baselines should be included for a more comprehensive evaluation. e.g. using prompts to guide the model to avoid bias in generated outputs,